# Choose Your Own Adventure: Using Twine for Gamified Interactive Learning in Veterinary Anaesthesia [note 1]

**DOI:** 10.3390/vetsci12020156

**Published:** 2025-02-11

**Authors:** José I. Redondo, M. Reyes Marti-Scharfhausen, Agustín Martínez-Albiñana, Ariel Cañón-Pérez, Álvaro J. Gutiérrez-Bautista, Jaime Viscasillas, E. Zoe Hernández-Magaña

**Affiliations:** 1Departamento de Medicina y Cirugía Animal, Facultad de Veterinaria, Universidad Cardenal Herrera-CEU, CEU Universities, C/Tirant lo Blanch, 7, 46115 Alfara del Patriarca, Valencia, Spain; eva.hernandezmagana@uchceu.es; 2AniCura Indautxu Hospital Veterinario, Bekoetxebarri Bidea, 5F, 48950 Erandio, Bizkaia, Spain; reyesvet92@gmail.com; 3AniCura Aitana Hospital Veterinario, C/de Xirivella, 16, 46920 Mislata, Valencia, Spain; agustin.martinez.vet@gmail.com; 4Experimental Surgery Unit, Institut de Recerca, Hospital Vall d’Hebron, VHIR Edifici Mediterrània, Pg. de la Vall d’Hebron, 129, Horta-Guinardó, 08035 Barcelona, Barcelona, Spain; arielcanonp@gmail.com; 5The Royal (Dick) School of Veterinary Studies, The University of Edinburgh, Easter Bush Campus, Midlothian EH25 9RG, UK; gutierrezbautistaalvarojesus@gmail.com; 6AniCura Valencia Sur Hospital Veterinario, Dirección: Av. de Picassent, 28, 46460 Silla, Valencia, Spain; jaimeviscasillas2@gmail.com

**Keywords:** gamification, veterinary education, simulation, experiential learning, decision-making skills

## Abstract

Veterinary anaesthesia requires theoretical knowledge and quick decision-making skills, but traditional methods often fall short in realistic training. To address this, we developed a web-based interactive learning system using Twine, featuring five clinical cases that simulate real-world anaesthesia scenarios with gamified elements like scoring and resource management. After a workshop for veterinary students, we surveyed 367 participants, of whom 90.8% found the system effective, and 97% agreed it enhanced their knowledge. The platform was deemed user-friendly, with 96.7% rating the workshop as “good” or “excellent”. Suggestions for improved mobile compatibility and additional resources were noted. Twine’s interactive approach reduces reliance on live animals and is globally accessible via web browsers. Future studies will assess its impact on skill retention and adaptability in various learning environments, providing a modern solution for veterinary education.

## 1. Introduction

Veterinary anaesthesia is a demanding discipline that requires extensive theoretical knowledge and the ability to make critical, real-time decisions in complex, high-pressure scenarios. Traditional educational approaches, such as lectures and case-based discussions, often focus on foundational knowledge but may fall short in preparing students for the quick thinking and adaptability essential to anaesthesia practice. The gap between theoretical understanding and hands-on clinical application calls for innovative educational tools that simulate real-world clinical complexity [1].

Over recent years, simulation-based learning has emerged as a powerful strategy for developing practical skills under controlled, realistic conditions. Simulation technologies allow students to learn by doing, providing opportunities to make decisions, experience consequences, and engage in reflective learning without the risk associated with live clinical practice. This approach has proven particularly useful in fields like anaesthesia, where students must master protocols, assess risks, and respond to complications accurately and confidently. Simulation-based learning effectively trains veterinary students in various fields beyond anaesthesia. For example, virtual simulations have enhanced surgical skills without risking live animals [2]. Simulations have been particularly effective in improving clinical reasoning skills [3], boosting student confidence and performance. Studies have shown that simulation-based learning can enhance knowledge retention, critical thinking, and decision-making skills by offering an immersive and active learning experience [1,4].

In parallel with these developments, gamified learning platforms have gained traction to increase student engagement and motivation. Gamification, incorporating game elements such as rewards, feedback, and interactive narratives, transforms traditional educational content into an active learning experience. Research has shown that gamified learning environments are particularly effective in sustaining attention, promoting deeper learning, and encouraging students to approach complex material with curiosity and perseverance [5]. Such approaches are well suited for anaesthesia training, where the stakes are high and the need for prompt, accurate responses is paramount.

Twine, an open-source tool for creating interactive, branching narratives, has emerged as a promising educational platform for designing immersive, decision-based learning experiences. Initially developed for storytelling [6], Twine has been successfully adapted for academic use in medicine, particularly in virtual patient (VP) simulations [7]. By creating a structured, narrative-driven learning environment, Twine enables students to navigate realistic clinical cases, make diagnostic and therapeutic choices, and observe the outcomes of their decisions. Studies on VP simulations suggest that they can significantly enhance clinical reasoning and knowledge acquisition, sometimes even outperforming traditional instructional methods in developing practical decision-making skills [4]. To our knowledge, Twine has been primarily used for human clinical cases, and its application in veterinary education, particularly anaesthesia, is novel and unexplored.

Twine’s interactive approach supports experiential learning theories like Kolb’s Learning Cycle, which emphasises active engagement with realistic scenarios for meaningful learning [8]. By incorporating gamification elements, Twine-based cases enhance the learning experience and boost student engagement with clinical content. This method encourages students to analyse, make decisions, and reflect on outcomes, strengthening critical thinking and practical skills for real-time decision-making.

Although gamified simulations are increasingly utilised in medical education, their adoption in veterinary training, particularly in anaesthesia, remains limited. This study evaluates the acceptance and perceived educational value of an interactive Twine-based module as a teaching tool in veterinary anaesthesia. The study explores the module’s potential to enhance experiential learning, critical thinking, and clinical decision-making skills by assessing veterinary students’ engagement and perceptions. We hypothesise that students will receive the Twine-based module well and provide an effective supplementary tool to traditional educational methods, fostering deeper learning and practical skill development in clinical training.

## 2. Materials and Methods

To create an interactive learning experience, we developed five clinical cases using Twine (version 2.8.1). Each case represented a realistic veterinary anaesthesia scenario that required critical decision-making at crucial points. For instance, students could select various anaesthetic management options and immediately see the outcomes of their choices, allowing for learning through trial and error in a safe, simulated environment (Figure 1). The cases covered essential principles of small animal anaesthesia, including patient assessment, stabilisation, anaesthetic protocol selection, monitoring, and complication management. Gamification elements, such as a points system for correct choices and a “budget” to simulate resource management constraints, were incorporated to enhance engagement and simulate real-world decision-making conditions. For example, one case focused on conducting a pre-anaesthetic evaluation for a dog scheduled for an ovariohysterectomy, requiring students to select an appropriate anaesthetic protocol and calculate precise drug dosages. Another case involved managing the anaesthesia of a dog with an intestinal obstruction, emphasising stabilisation and intraoperative monitoring. The series also included a scenario addressing the complexities of anaesthetising a dog with gastric dilatation–volvulus, a critical condition requiring prompt intervention. Additionally, students were presented with a cat suffering from multiple traumas, including pneumothorax, after a fall from a third floor, allowing them to apply concepts of stabilisation and anaesthetic management in emergencies. Finally, the cases included a cat diagnosed with feline lower urinary tract disease (FLUTD), highlighting the importance of careful anaesthetic planning in patients with obstructive conditions. These cases were hosted on itch.io, an independent online platform commonly used by game developers. Thus, they were accessible to students on any internet-enabled device [9].

This interactive Twine module was incorporated into a workshop within the Degree in Veterinary Sciences programme at Universidad Cardenal Herrera CEU (Valencia, Spain). Second, third, and fourth-year veterinary students were invited to participate. The workshop was designed to supplement existing coursework, providing students with an interactive, case-based learning experience. The Twine module was incorporated during the workshop as a hands-on component following a brief introductory lecture. Following the workshop, students were asked to complete an anonymous survey in Spanish to assess their knowledge and perceptions regarding the Twine-based module. The university’s Ethical Committee for Human Research approved this survey (CEEI 24/579), ensuring ethical standards were met.

The survey, conducted through Microsoft Forms, included quantitative and qualitative components to provide an overview of student feedback (Appendix A). It also collected demographic and career information, including questions regarding the specific course year, the student’s gender, and career aspirations within veterinary medicine.

The quantitative section used a five-point Likert scale to assess several aspects of the workshop. The survey consisted of questions designed to evaluate the students’ experience and satisfaction with an interactive veterinary anaesthesia workshop using Twine. Students were asked to rate their current interest in anaesthesia (INTEREST), the extent to which the workshop contributed to their training (CONTRIBUTION), and the effectiveness of using interactive clinical cases to enhance their knowledge (USEFULNESS). Additionally, the survey gauged the ease of use of the Twine platform (EASE) and overall satisfaction with the workshop (OVERALL). Open-ended questions allowed students to offer suggestions for improvement and share additional feedback, helping identify potential areas for future development. These metrics provided measurable insights into students’ perceptions of the module’s impact on their training. To enrich the data, the survey also included open-ended questions that invited students to provide qualitative feedback (FEEDBACK). These questions allowed students to suggest potential improvements (IMPROVEMENT) and share additional comments, adding depth to our understanding of their experiences with the Twine-based learning approach.

### Data Analysis

The statistical analysis used the R language program (version 4.4.2). Quantitative survey responses were analysed using descriptive statistics to summarise demographic data and satisfaction ratings. Likert scales assessed various aspects of the Twine platform, including ease of use, educational impact, and overall satisfaction.

A hybrid approach combining AI-assisted and human-led thematic analysis was employed for the qualitative analysis to process and categorise open-ended feedback efficiently. ChatGPT-4o [10] was selected for initial theme identification due to its ability to handle large volumes of text and accurately identify recurring patterns. Following the AI-based categorisation, human researchers reviewed and refined the themes to ensure accuracy, depth, and sensitivity to nuances in the data. This human oversight was essential for validating the AI-generated themes and addressing any subtleties in student feedback that might require additional context. Key themes, including “Interactivity and Engagement”, “Educational Value”, and “Platform Usability”, were finalised through this process, capturing a balanced view of students’ perceptions and highlighting areas for potential improvement. To visualise the qualitative data, word clouds were generated to represent frequently occurring terms, offering a visual summary.

## 3. Results

Of the 849 students invited to participate, 367 completed the survey, resulting in a response rate of 42%. The demographic profile indicates that most participants were enrolled in the fourth year (52.6%), followed by those in the third year (36.2%) and a smaller group from the second year (11.2%). Most respondents were female (77.4%), while males accounted for 20.4%, and 2.2% chose not to disclose their gender. In terms of specialisation, more than half focused on Small Animal Clinics (56.9%), with others specialising in Large Animal Clinics (15.8%), Animal Production (8.4%), Food Science and Technology (1.1%), and other fields (17.7%).

Feedback on Twine’s interactive and narrative format was overwhelmingly positive. In total, 52.9% of students agreed, and 37.9% strongly agreed, that the workshop contributed positively to their training in veterinary anaesthesia. Additionally, 21.5% of students agreed, and 75.5% strongly agreed, that the interactive cases improved their knowledge of anaesthesia concepts.

The Twine platform’s user-friendliness was also highly rated, with 18.3% of respondents rating the system as “easy” and 76.3% as “very easy”. Regarding overall satisfaction, 29.7% of students described the workshop as “good”, while 67.0% rated it as “excellent”, suggesting strong support for Twine’s effectiveness in this educational context. Figure 2 shows the Likert graphs for INTEREST, CONTRIBUTION, USEFULNESS, EASE, and OVERALL.

Qualitative feedback highlighted multiple positive aspects of the workshop. Many students found Twine’s interactivity engaging, citing the opportunity to apply theoretical knowledge to realistic scenarios as a key benefit. Students appreciated the detailed feedback for each decision, including explanations for correct and incorrect choices, which helped them learn from their mistakes. Furthermore, several students noted improvements in the workshop compared to previous years, specifically regarding graphical quality and content depth, and expressed gratitude for the instructors’ commitment. Figure 3 shows the word cloud of the feedback.

Students also offered constructive feedback on areas for improvement. Some encountered technical issues, such as images overlapping with text, which affected readability. Additionally, there were requests for supplementary resources, including summaries of completed cases and more in-depth explanations of specific anaesthesia concepts. A few students suggested optimising Twine for better mobile compatibility to improve smartphone accessibility.

## 4. Discussion

This study underscores Twine’s potential as a versatile, interactive educational tool for veterinary anaesthesia. Twine offers a valuable alternative to traditional learning methods by combining case-based learning with dynamic decision-making. Student feedback highlights Twine’s ability to enhance engagement, knowledge retention, and practical skills, which are essential in preparing students for the complexities of anaesthesia practice. These findings are consistent with research on simulation-based learning in veterinary and medical education, which has shown benefits in deepening conceptual understanding, promoting active participation, and improving clinical reasoning [1,4].

A key strength of Twine is its accessibility and ease of use, making it a practical tool for experiential learning in diverse educational settings. Grounded in Kolb’s experiential learning theory, Twine’s interactive, decision-based structure enables students to engage with clinical scenarios actively, make real-time decisions, and observe consequences, thereby simulating the realities of veterinary practice in a reflective, hands-on manner [8]. Similar findings from virtual patient (VP) simulation studies suggest that practice-based learning platforms can effectively bridge theoretical knowledge and practical skills essential for decision-making in high-stakes clinical environments [4]. These simulations have demonstrated their effectiveness in enhancing clinical reasoning and decision-making abilities across diverse medical disciplines [11]. VP platforms provide several advantages for medical education. They enable students to practice repeatedly without risking harm to actual patients; offer standardised tasks and completion criteria to ensure consistent learning experiences; and provide real-time feedback, boosting students’ confidence in their clinical skills [12,13].

Twine’s integration with itch.io (https://itch.io/) (accessed on 2 December 2024) offers another advantage: cross-device access allows students to engage with the material independently. This flexibility aligns with universal design principles in education, which advocate for resources adaptable to diverse learner needs [14]. Moreover, Twine’s compatibility with widely used web browsers featuring built-in translation capabilities supports a multilingual student body, enhancing inclusivity and accessibility for international veterinary students [15]. This adaptability makes Twine a potentially valuable resource for global veterinary education, accommodating students from varied linguistic and cultural backgrounds.

The platform also supports autonomous learning, empowering students to navigate clinical scenarios and make educational decisions independently. This self-directed learning feature is particularly relevant in clinical fields, where ongoing professional development is crucial [16]. During the COVID-19 pandemic, this Twine tool provided an essential means of continuing practical education remotely in our university, enabling students to progress independently when in-person training was limited. This adaptability highlights Twine’s value as a tool for promoting lifelong learning skills and preparing students to take responsibility for their clinical education.

In addition to its practical benefits, Twine aligns with ethical practices in veterinary education. By simulating clinical scenarios without live animals, Twine offers a humane training alternative, reducing reliance on live animals in early training stages. This aligns with modern ethical standards in veterinary training, offering students a preparatory learning phase that supports animal welfare and student confidence-building before engaging in real-world clinical practice [17].

While Twine’s strengths are notable, the study identified areas for improvement. Technical issues, such as overlapping text and images, impacted user experience, highlighting the importance of stability and usability in educational design. Optimising Twine for mobile device compatibility would address these issues and enhance accessibility, especially given the widespread reliance on smartphones in professional learning contexts. Addressing these usability challenges in future versions could ensure a seamless device experience.

Students also indicated a need for supplementary resources to enhance their understanding of complex concepts. Suggestions included case summaries and additional explanations of specific anaesthesia principles. Research on multimodal learning supports the integration of interactive modules with supplementary materials, which can reinforce comprehension and retention in technical fields [14]. Future Twine modules could incorporate these resources—such as detailed decision feedback, follow-up quizzes, or case summaries—to create a more comprehensive learning experience.

The AI tool facilitated a rapid, systematic extraction of common themes from open-ended responses, streamlining the initial stages of analysis and laying a foundation for further categorisation. This hybrid AI–human approach effectively identified critical themes in student feedback, combining the speed of AI categorisation with human review to preserve the depth and context of responses. Recent studies support the integration of AI tools like ChatGPT-4 in qualitative research, highlighting their efficiency in identifying recurring patterns and streamlining initial theme identification [18,19]. This dual approach is particularly beneficial for studies with large qualitative datasets, enhancing both efficiency and depth in thematic analysis. However, AI’s limitations in interpreting nuanced feedback should be considered, and human oversight is essential to ensure balanced and accurate insights.

Simulation-based and gamified learning tools in veterinary and medical education significantly enhance student engagement, knowledge retention, technical skills, and clinical decision-making while reducing the need for live animals. Examples like the TARGET ultrasound simulator [20], the interactive Neuropathology iBook [21], and virtual reality (VR) simulators for tasks like canine intubation [22] show their effectiveness in skill development and confidence building in low-stakes settings. Tools such as SimuVet for epidural anaesthesia training [23] and GASMAN for anaesthetic pharmacology [24] further deepen understanding and support real-time decision-making.

Several interactive and digital platforms have been successfully utilised in veterinary and medical education to enhance student learning. For example, the CASUS system has been implemented in veterinary medicine with positive feedback [25]. During the COVID-19 pandemic, Virtual Problem-Based Learning (v-PBL) provided online access to real clinical cases, though students felt it could not fully replace hands-on training [26]. In veterinary anatomy education, augmented reality (AR) tools created interactive 3D models using canine CT and MRI scans, aiding in understanding complex subjects like neuroanatomy [27]. In human medicine, VR and AR have been widely adopted for simulation-based training and immersive learning experiences, with the emerging concept of the metaverse offering innovative possibilities [28]. These examples illustrate the significant potential of interactive platforms to augment the educational experience; however, their efficacy is contingent upon their thoughtful integration and the recognition of their role as supplementary tools alongside conventional clinical training.

This study has several limitations that should be considered when interpreting the findings. Participation in the survey was voluntary, which may have introduced self-selection bias. It is possible that the students who responded were either those most motivated or, conversely, those who found the tool less engaging, leading to a skewed representation of the overall student experience. Additionally, the study focused on a single institution, which limits the generalisability of the results to other educational contexts. Another area for improvement is the inability to directly measure the impact of the Twine module on bridging the gap between theoretical knowledge and practical skills, as this would require longitudinal studies or controlled comparisons with other educational methods. Future research could address these gaps by implementing mandatory participation, expanding the survey to multiple institutions, and employing methodologies that assess the long-term retention of clinical skills and knowledge gained through Twine. Such studies could also explore how Twine compares to other simulation-based tools in enhancing experiential learning and decision-making in veterinary education.

## 5. Conclusions

In summary, Twine demonstrates significant potential as a flexible, accessible, and humane educational tool that meets the evolving needs of veterinary education. Its ease of use, inclusivity, and alignment with ethical training practices make it an attractive solution for experiential learning in veterinary anaesthesia. Addressing technical challenges and incorporating supplementary resources could enhance Twine’s educational impact. Future research may explore Twine’s broader applications across other clinical specialisations and assess its role in supporting technology-enhanced, self-directed learning in veterinary and medical education.

## Figures and Tables

**Figure 1 vetsci-12-00156-f001:**
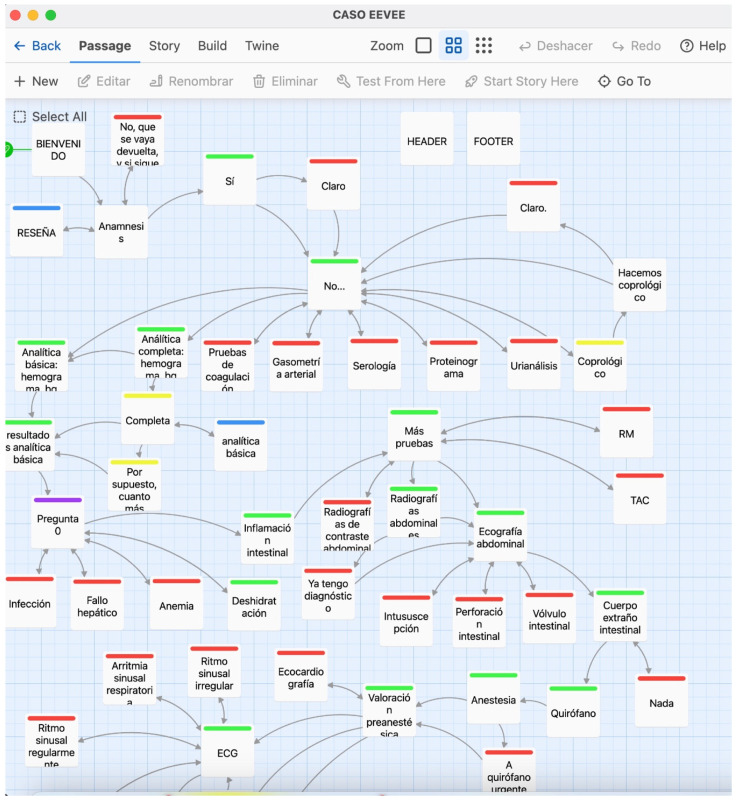
Screenshot of the Twine interface showing the interactive narrative structure of a clinical case. Each box, or ”passage”, represents a decision point or informational node. Arrows represent the connection between passages. Colours indicate the nature of each choice: green for correct decisions, red for incorrect choices, yellow for neutral options, and blue for informational passages.

**Figure 2 vetsci-12-00156-f002:**
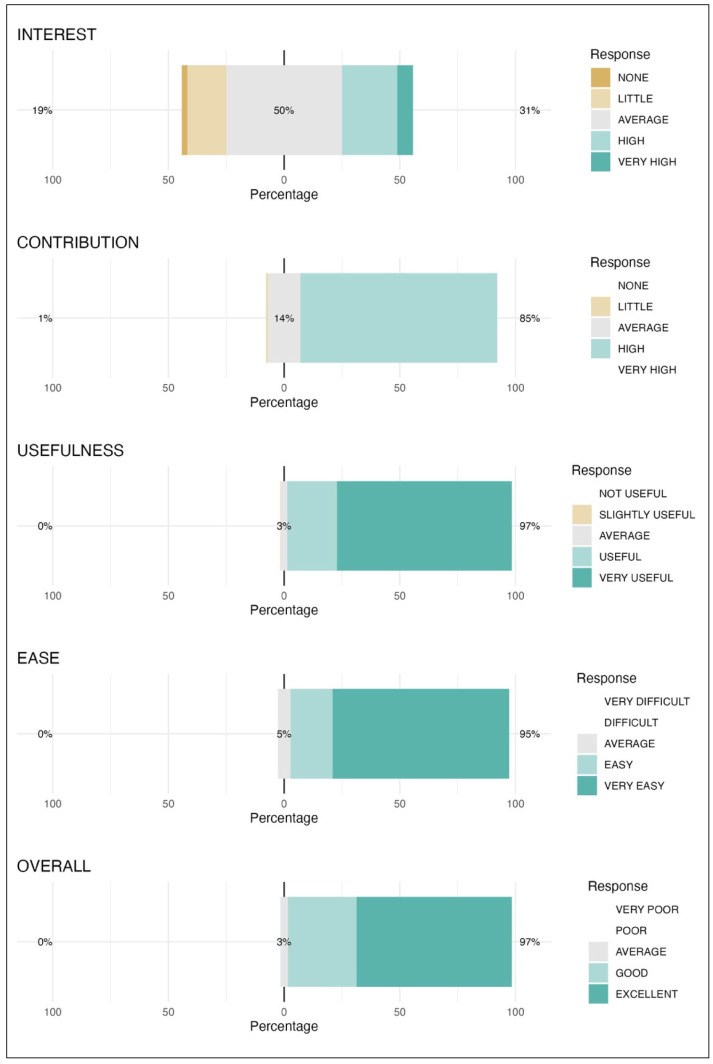
Likert scale responses for students’ perceptions of the Twine-based veterinary anaesthesia workshop. The categories assessed were INTEREST in anaesthesia, perceived CONTRIBUTION to training, USEFULNESS of interactive clinical cases, EASE of using the Twine platform, and OVERALL satisfaction. The distribution of responses is shown as percentages, with colour-coded categories representing levels of agreement or satisfaction, ranging from “None” to “Very High” or equivalent terms, based on the specific question.

**Figure 3 vetsci-12-00156-f003:**
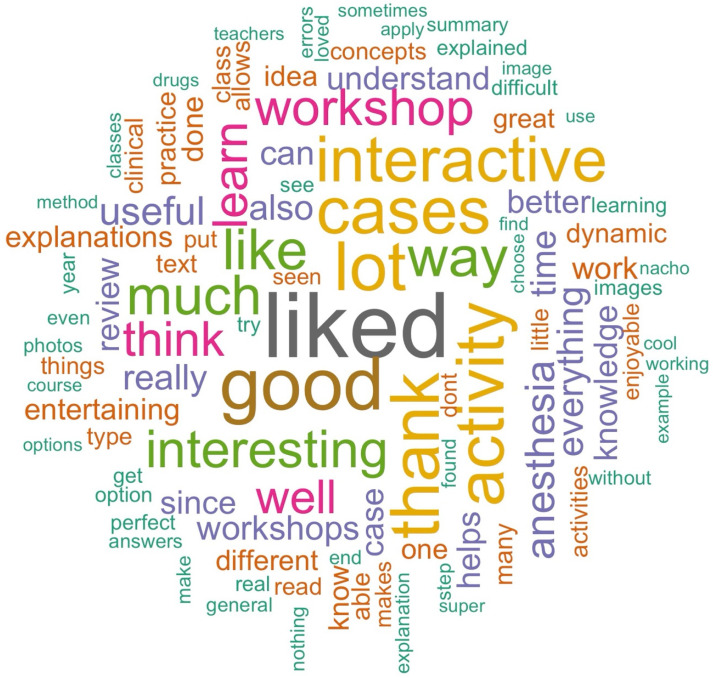
Word cloud of student feedback. The size of each word reflects the frequency with which it was mentioned by students in their responses, highlighting key themes and perceptions of the Twine-based workshop.

## Data Availability

The data supporting this study’s findings are available from the corresponding author upon reasonable request.

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
