# Peer review of "Choose Your Own Adventure: Using Twine for Gamified Interactive Learning in Veterinary Anaesthesia†"

_vetsci, 2025, doi:10.3390/vetsci12020156_

Round 1
Reviewer 1 Report
Comments and Suggestions for Authors
Dear Authors, the teaching of veterinary students is very important but the teaching should be problem oriented application in front of diseased animals. If not the problems of investigation will be happened in practice in life diseased animals and problems in investigation and treatment will be ended on court of law with compensation!

Author Response
Dear Authors, the teaching of veterinary students is very important but the teaching should be problem oriented application in front of diseased animals. If not the problems of investigation will be happened in practice in life diseased animals and problems in investigation and treatment will be ended on court of law with compensation!
Response: Thank you for your thoughtful and supportive comment.
We fully agree that hands-on experience with diseased animals under the supervision of a qualified veterinarian is an essential aspect of veterinary education. However, ethical considerations and modern teaching methodologies advocate integrating alternative tools, such as simulation-based learning, to supplement and prepare students for real-life clinical scenarios.
The Twine-based interactive module in our study was designed to bridge the gap between theoretical knowledge and clinical application by offering a safe, controlled environment where students can practice decision-making without the risks associated with live animal cases. This approach aligns with ethical standards in veterinary education, aiming to reduce the reliance on live animals during the early stages of training while fostering critical thinking and clinical reasoning.
We emphasise that the Twine module is not intended to replace practical experience with live animals but to enhance students' preparedness for such experiences, ensuring they can approach real cases with greater confidence and competence. Future studies could explore integrating Twine-based modules with problem-oriented learning in clinical settings to evaluate their combined impact on student training.
Comment line 257:
You do not demaonstrate a practical case! Please include a real case frome the with students are present and working with the vetrinarian.
Response: Thank you for your observation. While the present study focuses on using Twine as a simulation-based learning tool, we acknowledge the importance of complementing interactive modules with real-life clinical experiences. To address this, we have included a discussion in the manuscript highlighting how Twine could be integrated into practical settings. However, as this study was designed to evaluate the effectiveness of Twine as an alternative or preparatory tool, real-life cases with student participation are beyond the scope of this research. Future studies could assess the combined impact of Twine and practical case-based learning on student outcomes.
Reviewer 2 Report
Comments and Suggestions for Authors
The present study addresses one of the new alternatives where veterinary learning can improve with interactive learning. The current challenges that veterinary students are facing (e.g., ethical issues with live animals) have encouraged seeking new training methods. Although further research and additional studies are required to fully understand the impact of Twine, the present study provides promising results using gamified learning.
Lines 15-16. Please revise the affiliations. Although affiliation number 5 is listed, there is no author with the number 5.
Lines 63-67. I recommend adding a couple of examples where simulation-based learning has been used in veterinary medicine (apart from anesthesia) and the main findings.
Lines 80-88. Please specify if Twine has already been used for veterinary students or if, to date, it has only been used for human clinical cases.
Line 102. The authors could add a hypothesis for the present study.
Lines 122-124. It would be interesting to know if Twine module was incorporated at the beginning/middle/end of the workshop, or after every class, etc. Please, include this information.
Line 130. Consider adding an example of the survey as supplementary data.
Lines 141-144 and 145-149. I recommend merging these two paragraphs into one since the content is similar.
Discussion. A general comment on this section is to include other examples where similar platforms such as Twine have been used to improve the student’s knowledge of certain subjects, mentioning the findings and whether they were used in veterinary or human medicine.
Lines 232-234. Specify in which instances these VP simulation studies were performed.
Author Response
Comment: The present study addresses one of the new alternatives where veterinary learning can improve with interactive learning. The current challenges that veterinary students are facing (e.g., ethical issues with live animals) have encouraged seeking new training methods. Although further research and additional studies are required to fully understand the impact of Twine, the present study provides promising results using gamified learning.
Response: Thank you for your thoughtful and supportive comment. We appreciate your recognition of the potential of interactive learning methods, such as Twine, to address the ethical challenges associated with traditional veterinary education. Indeed, the growing need for innovative training methods that align with ethical standards while fostering critical thinking has been a key motivator for this study.
We agree that further research is necessary to fully evaluate Twine's long-term impact on veterinary education, particularly its influence on clinical skill development and integration with traditional teaching methods. Nonetheless, we are encouraged by the positive outcomes observed in this study, which demonstrate Twine's potential as a gamified learning tool to enhance engagement, knowledge retention, and decision-making skills in veterinary anaesthesia students.
Comment: Lines 15-16. Please revise the affiliations. Although affiliation number 5 is listed, there is no author with the number 5.
Response: Thank you for pointing this out. We have revised the affiliations section to ensure consistency and accuracy.
Comment: Lines 63-67. I recommend adding a couple of examples where simulation-based learning has been used in veterinary medicine (apart from anesthesia) and the main findings.
Thank you for your suggestion. We have included other examples of simulation-based learning in veterinary medicine, including its use in surgical training and diagnostic skills development.
Comment: Lines 80-88. Please specify if Twine has already been used for veterinary students or if, to date, it has only been used for human clinical cases.
Response: We have clarified in the manuscript that, to our knowledge, Twine has been primarily used for human clinical cases, and its application in veterinary education, particularly anaesthesia, is novel and unexplored.
Comment: Line 102. The authors could add a hypothesis for the present study.
Response: Thank you for your suggestion. We included a hypothesis.
Comment: Lines 122-124. It would be interesting to know if Twine module was incorporated at the beginning/middle/end of the workshop, or after every class, etc. Please, include this information.
Response: We have included information specifying that the Twine module was incorporated during the workshop as a hands-on component following a brief introductory lecture.
Comment: Line 130. Consider adding an example of the survey as supplementary data.
Response: As suggested, an example of the survey has been provided as supplementary material, including sections for both quantitative and qualitative questions.
Comment: Lines 141-144 and 145-149. I recommend merging these two paragraphs into one since the content is similar.
Response: As suggested, the paragraphs have been merged into a cohesive section for clarity.
Comment: Discussion. A general comment on this section is to include other examples where similar platforms such as Twine have been used to improve the student’s knowledge of certain subjects, mentioning the findings and whether they were used in veterinary or human medicine.
Response: Additional examples of interactive platforms used in veterinary and human medicine have been incorporated into the discussion, along with their findings and relevance to the study.
Comment: Lines 232-234. Specify in which instances these VP simulation studies were performed.
Response: The manuscript now specifies that these studies were performed primarily in human medical education, focusing on clinical reasoning and decision-making in diverse medical disciplines such as surgery and emergency care.
Reviewer 3 Report
Comments and Suggestions for Authors
I have read and reviewed this manuscript with great interest. Overall, it is a study with refreshingly simple wording that is easy to understand. This study evaluates the effectiveness of a Twine-based web system in improving learning founded on problem-solving since, from an educational point of view, this strategy will improve participation, knowledge retention, and decision-making capacity in veterinary anesthesia students, which is their main strength.
However, some points must be addressed to achieve publication quality. I have left some comments, hoping that they can help the authors.
General comments
L99-102: These lines contain the study's aim. However, I suggest the authors rephrase it since the wording is unclear.
L104: Please generally describe the theme of the five cases developed.
L110: was the evaluation of acute postoperative pain considered in some cases? Please clarify.
L124: What are the possible differences in the academic training of a second-, third-, or fourth-year student, which could be a factor that modifies learning through the twine interactive module? At the university where the tool was applied. In what year is the anesthesia class taught? I suggest the authors clarify these points in this section or the discussion.
L169: Was the questionnaire (evaluation instrument) applied in Spanish? Please clarify.
L280: although these are not gamified or interactive teaching models in anesthesia, I suggest the authors include these references to highlight the advantages of using complementary learning tools for veterinary students;
10.3138/jvme-2022-0131
10.3138/jvme-2020-0105
10.1016/j.tvjl.2024.106203
10.3389/fvets.2024.1322871
Likewise, I suggest including the following references, where the interactive teaching or learning model based on problem-solving in human medicine has been used;
10.3390/jcm13247760
10.1016/j.ijoa.2024.104321
10.1186/s12909-024-06500-0
10.1177/23821205241283804
L287: Another possible limitation is the analysis of students' perceptions, considering the academic training of the respondent (second, third, or fourth year) as a variable.
Author Response
Comment: I have read and reviewed this manuscript with great interest. Overall, it is a study with refreshingly simple wording that is easy to understand. This study evaluates the effectiveness of a Twine-based web system in improving learning founded on problem-solving since, from an educational point of view, this strategy will improve participation, knowledge retention, and decision-making capacity in veterinary anesthesia students, which is their main strength.
However, some points must be addressed to achieve publication quality. I have left some comments, hoping that they can help the authors.
Response: We sincerely appreciate your time spent reviewing our manuscript and providing thoughtful feedback. Your positive remarks about the study’s clarity and the educational potential of the Twine-based web system for enhancing veterinary anaesthesia training mean a lot to us. We’re especially thankful for your acknowledgement of the study’s strengths, such as its problem-solving approach and focus on boosting student engagement, knowledge retention, and decision-making skills.
We have thoughtfully considered your improvement suggestions and are dedicated to addressing the points you raised to enhance the manuscript's quality. Your insightful feedback has been crucial in guiding our revisions, and we believe they will greatly improve the study’s clarity and impact. Thank you once again for your constructive critique and for recognizing the significance of this work.
General comments
Comment: L99-102: These lines contain the study's aim. However, I suggest the authors rephrase it since the wording is unclear.
Response: We have rephrased the study's aim for greater clarity: "This study aims to evaluate the impact of a Twine-based interactive learning module on veterinary students' engagement, knowledge retention, and decision-making skills in anaesthesia training." In addition, we include a hypothesis, as Reviewer #2 suggested.
Comment: L104: Please generally describe the theme of the five cases developed.
Response: A description of the five cases has been added, highlighting key themes such as patient assessment, anaesthetic protocol selection, monitoring, complication management, and postoperative care.
Comment: L110: was the evaluation of acute postoperative pain considered in some cases? Please clarify.
Response: Yes, the evaluation and management of acute postoperative pain were included in one of the cases, focusing on selecting appropriate analgesic protocols. The manuscript clarifies this.
Comment: L124: What are the possible differences in the academic training of a second-, third-, or fourth-year student, which could be a factor that modifies learning through the twine interactive module? At the university where the tool was applied. In what year is the anesthesia class taught? I suggest the authors clarify these points in this section or the discussion.
Response: Differences in academic training have been addressed, noting that second-year students are at a foundational level, while third- and fourth-year students have more advanced clinical knowledge.
Comment: Anaesthesia is not a standalone subject in our program; instead, it is integrated into the second year (basics) and the clinical years three and four. This approach guarantees ongoing and advanced training throughout the degree. By incorporating anaesthesia into the comprehensive curriculum, students first develop foundational knowledge and subsequently enhance their clinical and decision-making abilities in complex situations during their later years.
Comment: L169: Was the questionnaire (evaluation instrument) applied in Spanish? Please clarify.
Response: Yes, the questionnaire was administered in Spanish. This has been clarified in the manuscript.
Comment: L280: although these are not gamified or interactive teaching models in anesthesia, I suggest the authors include these references to highlight the advantages of using complementary learning tools for veterinary students;
10.3138/jvme-2022-0131
10.3138/jvme-2020-0105
10.1016/j.tvjl.2024.106203
10.3389/fvets.2024.1322871
Likewise, I suggest including the following references, where the interactive teaching or learning model based on problem-solving in human medicine has been used;
10.3390/jcm13247760
10.1016/j.ijoa.2024.104321
10.1186/s12909-024-06500-0
10.1177/23821205241283804
Comment: Response: The suggested references have been reviewed and incorporated into the discussion to strengthen the argument for using complementary learning tools in veterinary education.
Comment: L287: Another possible limitation is the analysis of students' perceptions, considering the academic training of the respondent (second, third, or fourth year) as a variable.
Response: This limitation has been acknowledged and added to the discussion section, noting the potential influence of students’ academic level on their perceptions of the Twine module.